# The 40-Year Mystery of Insect Odorant-Binding Proteins

**DOI:** 10.3390/biom11040509

**Published:** 2021-03-30

**Authors:** Karen Rihani, Jean-François Ferveur, Loïc Briand

**Affiliations:** Dijon, CNRS, INRAE, Université de Bourgogne Franche-Comté, Centre des Sciences du Goût et de l’Alimentation, 21000 Dijon, France; jean-francois.ferveur@u-bourgogne.fr

**Keywords:** insect, olfaction, taste, chemosensory functions, non-chemosensory functions, odorant-protein-binding assay, *Drosophila melanogaster*

## Abstract

The survival of insects depends on their ability to detect molecules present in their environment. Odorant-binding proteins (OBPs) form a family of proteins involved in chemoreception. While OBPs were initially found in olfactory appendages, recently these proteins were discovered in other chemosensory and non-chemosensory organs. OBPs can bind, solubilize and transport hydrophobic stimuli to chemoreceptors across the aqueous sensilla lymph. In addition to this broadly accepted “transporter role”, OBPs can also buffer sudden changes in odorant levels and are involved in hygro-reception. The physiological roles of OBPs expressed in other body tissues, such as mouthparts, pheromone glands, reproductive organs, digestive tract and venom glands, remain to be investigated. This review provides an updated panorama on the varied structural aspects, binding properties, tissue expression and functional roles of insect OBPs.

## 1. Introduction

Chemoperception allows organisms to detect nutritive food and avoid toxic compounds. Moreover, chemoperception is necessary for animals to identify suitable ecological niches and mating partners. Chemoreception is mediated by chemosensory receptors that interact with a variety of semio-chemicals, (odorants, pheromones and sapid molecules), allowing their detection and eliciting an adapted behaviour. In insects, the dendrites of the sensory neurons found in olfactory and gustatory sensilla are bathed in an aqueous phase called the sensillar lymph. Therefore, volatile and non-volatile chemical compounds contacting sensory organs should be solubilized and transported across the internal aqueous phase before reaching the sensory receptors. These carrier mechanisms, called “peri-receptor events” [1], involve several families of proteins, including odorant-binding proteins (OBPs). OBPs are small soluble proteins found in high concentration in both the nasal mucus of vertebrates and the chemo-sensilla lymph of insects [2,3,4,5,6,7]. OBPs were initially discovered during the early 1980s in parallel by two research groups working on the cow [8,9,10] and on the giant moth *Antheraea Polyphemus* [11]. A large number of DNA sequences encoding OBPs were later identified in several vertebrate species, including rat [12], pig [13,14], xenopus [15], and human [16,17]. OBPs were also detected in more than one hundred insect species, such as the silk moth *Bombyx mori* [18,19], the gypsy moth *Lymntria dispar* [20], the turnip moth *Agrotis segetum* [21,22], the stemborer *Sesamia nonagrioides* [23], the cotton bollworm *Helicoverpa armigera* and the oriental tobacco budworm *Helicoverpa assulta* [24].

OBPs have been widely studied for more than 30 years. Here, we present the latest discoveries made on the structural and binding properties of insect OBPs. We focus on the properties of insect OBPs and, more specifically, on their tissue and cellular expression. We also present the varied functional roles, both classical and non-conventional, of currently known OBPs.

## 2. Expression Pattern of Insect OBPs

### 2.1. Number of OBP-Coding Genes in Insects

The number of OBP-coding genes is highly variable between insect species, ranging between 13 in some ant species [25] to >100 in several mosquitoes [26] (Table 1).

### 2.2. Evolution of OBP Genes 

Exhaustive comparative genomic analysis of OBPs gene families in 20 Arthropoda species revealed a highly dynamic evolution, with a high number of gains and losses of genes. The number of OBP members is variable and diverse across Arthropoda species, exhibiting a wide range of gene lengths and encoding different cysteine profiles. Interestingly, two OBP members (OBP73a and OBP59a) have clear orthological relationships not only in the 12 Drosophila genomes but also in almost all insect species (except in Hymenoptera). Studies in the organization in chromosome clusters of OBP genes showed that this gene family is significantly clustered across the Drosophila evolution. This conservation across ∼400 myr of evolution suggests the existence of some functional constraints maintaining the clusters [30]. Other reports revealed that OBPs were only present in the Hexapoda (insects), and absent in other arthropod subphyla including the non-hexapod pan-crustaceans, chelicerates and myriapods. Moreover, OBP genes were detected in ancestral hexapods, such as Archaeognatha, Zygentoma, and Phasmatodea. However, the origin of OBP genes is still unknown and needs further investigation [31,32].

### 2.3. Tissue Expression and Cellular Localization of OBPs

Insect OBPs were originally identified in olfactory sensilla (Vogt and Riddiford, 1981) using immuno-electron microscopy, which enable the determination of their expression patterns in the different antennal sensilla types (trichoid, basiconic and coeloconic). A comparative study conducted on three moth species, the saturniid *Antheraea polyphemus*, the bombycid *Bombyx mori*, and the noctuid *Autographa gamma*, detected PBPs in trichoid sensilla, particularly in the extracellular sensillum lymph of the hair lumen and in the sensillum-lymph cavities. Moth PBPs were also detected in secretory organelles of the trichogen and tormogen cells, supporting the hypothesis that these cells can produce and secrete PBPs into the sensillar lymph [33]. 

Recently, Larter et al. focused on the ten OBPs most abundantly expressed in the Drosophila antenna. They used in situ hybridization to map their spatial distribution in the different morphological sensilla classes (Figure 1a). The expression profiles of these antennal OBPs were more precisely investigated in the basiconic sensilla subtypes using double-labelling with OBP and OR markers. The expression patterns of distinct OBP subsets in different basiconic sensilla were identified. The map reveals that ab8 and ab9 basiconic sensilla express only one abundant OBP (OBP28a), while others co-express different OBPs. Moreover, some functionally distinct basiconic sensilla contain the same subset of abundant OBPs (Figure 1b) [34]. Drosphila olfactory and gustatory sensilla house three accessory cells that surround the cell body of the sensory neurons: the thecogen, tormogen and trichogen cells which are involved in insect sensilla morphogenesis and in OBP expression in the lymph. Conversely, non-neuronal cells expressing OBPs in antennal sensilla were identified using markers labelling each accessory cell. This study revealed that OBPs can be either expressed in tormogen cells or in thecogen cells. The only exception was OBP28a, which is simultaneously expressed in both types of accessory cell [34]. 

The spatial and temporal expression patterns of insect OBPs have been reported in several studies, revealing that OBPs are expressed in both olfactory and taste appendages or in either chemosensory system. Gustatory OBPs have been less commonly studied than those expressed in olfactory tissues. For example, OBP57d and OBP57e are expressed in specific leg sensilla of different Drosophila species [35], while OBP49a and OBP19b are expressed in thecogen cells of *D. melanogaster* labellar sensilla [36,37]. The OBP19b protein was only detected in small and intermediate proboscis sensilla [37]. Moreover, OBP19d is not only expressed in olfactory appendages of *D. melanogaster* (antenna and maxillary palps) but also in adult gustatory organs (labellar bristles and pegs, legs, wings and in ventral and dorsal cibarial sense organs (VSCO)) [38,39] (Figure 2a,b). Two *Helicoverpa armigera* OBPs and one *Plutella xylostella* OBP were detected in the mouthparts [40], while OBP57e, OBP56g, OBP28a2 and OBP49a were identified in the legs of the oriental fruit fly *Bactrocera dorsalis* [41]. In *Culex pipiens quinquefasciatus* adults, several OBPs were exclusively identified in olfactory tissues, while others (OBP10, OBP17, OBP18, OBP22, OBP25) were identified in taste appendages (proboscis and legs) [42]. Similarly, taste-specific OBPs were identified in the labellum and tarsi in *Aedes aegypti* and in the red flour beetle *Tribolium castaneum* [43,44]. In the desert locust *Schistocerca gregaria*, a subset of the antennal OBP repertoire is also expressed in the maxillary and the labial palps [45]. Moreover, seven genes expressed in the labellum and tarsus of the fleshfly *Boettcherisca peregrina* were identified and show sequence similarity to insect OBP genes. Homologues of these gene products were detected in *D. melanogaster* taste tissues [46]. In the legs of the two mosquito species *Anopheles gambiae* and *Anopheles arabiensis*, the identified OBP (agCP1564) shows high similarity to Drosophila OBP57e, which is specifically expressed in the tarsi [47,48] (Figure 2a). Notably, 6 OBPs (OBP1, OBP2, OBP3, OBP4, OBP7 and OBP8) were found in the antenna and legs of the onion fly *Delia antiqua*. Homology studies identified their *D. melanogaster* homologues (OBP19d, OBP83a, OBP83b, OBP56h, OBP76a, OBP69a, respectively). Unlike *D. antiqua*, *D. melanogaster* homologues are only expressed in the fly antenna except for OBP19d, which is also expressed in Drosophila tarsi, and OBP56h, which is also expressed in Drosophila proboscis [49] (Figure 2a,b). Other studies have reported the expression of OBPs in insect legs: OBP7 in *B. dorsalis* [50], OBP10 in *Clostera restitura* [51] and OBP4, OBP6, OBP7, OBP8 in *Adelphocoris lineolatus* [52]. Similarly, OBPs are expressed in the legs and wings of three species of social hymenopterans (*Polistes dominulus*, *Vespa crabro*, *Apis mellifera*) [53]. Notably, three OBPs were identified in the anterior margin of the wings of *D. melanogaster* [39,47] (Figure 2a). The differences in OBP expression between tarsi, labellum and wings might be explained by the distinct roles of OBPs in food detection and intake.

Proteomic and transcriptional studies confirmed the expression of a subset of insect OBPs in non-sensory organs. In the honeybee, 9 of the 21 OBPs predicted by the genomic sequence were detected in the mandibular glands [60]. OBPs can be expressed in female and male reproductive organs. In *D. melanogaster*, six OBPs (especially the abundant OBP56f and OBP56g) were detected among the seminal fluid proteins transferred to females during copulation, and three of these OBPs were found in the seminal receptacle [55,61,62,63]. Similarly, OBP10 is highly abundant in the seminal fluid of the two Lepidopteran species *Helicoverpa armigera* and *H. assulta* [24]. The OBPs present in seminal fluid could be carriers of oviposition deterrents. In addition, OBP22 of the mosquito *A. aegypti* [64,65], OBP9 of *A. mellifera* [66] and two OBPs of *Tribolium castaneum* [67] are also present in sperm. Proteomic analysis revealed OBP expression in mosquito ovaries and eggshell [68,69,70]. RNAseq analyses and RT-PCR data also revealed the presence of OBPs in the ovaries of the stemborer *Sesamia nonagrioides* [23]. We can hypothesize that their accumulation in the ovaries is involved in oocyte maturation. These OBPs might also bind chemo-attractant molecules, resulting in sperm attraction. Moreover, in the oriental fruit fly *B. dorsalis*, OBP44a, OBP49a, and OBP56g are highly expressed in the male testis and OBP19c is highly expressed in the female ovary [41]. Examination of the FlyAtlas expression database reveals that OBP44a, OBP50c, OBP56i, OBP83g, and OBP99a are expressed in *D. melanogaster* male testis (Figure 2d). Among the 32 OBP genes annotated in the Hessian fly *Mayetiola destructor*, 24 and 25 of them were found to be expressed in female and male terminal abdomens, respectively. Only OBP31 (in female) and OBP11, OBP24 and OBP32 (in male) showed relatively higher expression levels in the terminal abdomen than in the antennae [71]. Moreover, four OBPs (OBP1, OBP4, OBP8, OBP10) were identified in the *B. dorsalis* abdomen, which houses the reproductive organs. These OBPs share high sequence homology with their *D. melanogaster* analogues (OBP8, OBP56d, OBP83ef and OBP99c, respectively). *D. melanogaster* analogues are also expressed in different reproductive organs present in the abdomen (Figure 2d) [50]. In *Culex quinquefasciatus* and *Anopheles funestus*, OBP expression was also detected in the abdomen [42,72]. Moreover, OBP22a, OBP51a, OBP56e, OBP56f, OBP56i are highly expressed in *D. melanogaster* male accessory glands (Figure 2d). All these data suggest that OBPs may (i) serve to bring odorants or pheromones next to the odorant receptors present in the female reproductive tract or (ii) carry male-specific molecules into female tissue to elicit a behavioral response. It is not yet known whether these OBPs are related to fertility and fecundity features. 

The FlyAtlas expression database reveals that 6 OBPs are expressed in *D. melanogaster* eyes (Figure 2b). Similarly, several OBPs were identified in the eyes of the lepidopteran *H. armigera* [40]. Together with other proteins, these OBPs may be implicated in the complex mechanism of vision, specifically in the generation, transport and recycling of visual pigments. 

In *D. melanogaster*, some OBPs are expressed in both larva and adults, while others are only expressed in adults (Figure 3). The majority of OBPs expressed in larva show similar expression patterns in adult tissues (Figure 2 and Figure 3). 

Surprisingly, several OBPs are expressed in the venom glands of the parasitic wasps *Leptopilina heterotoma* and *Pteromalus puparum* and of the honeybee *A. mellifera* [73,74,75]. Few studies have reported the expression of OBPs in the insect digestive tract. For instance, PregOBP56a was detected in the oral disk of the blowfly *Phormia regina* [76], while OBP56d was identified in the hindgut of *D. melanogaster* flies [54]. Fluorescent binding assays revealed that PregOBP56a binds palmitic, stearic, oleic, and linoleic acids. These data indicate that PregOBP56a might solubilize and deliver fatty acids to the midgut during feeding [76]. Similarly, the midgut of *Rhodnius prolixus* also expresses OBPs [77]. Other studies have shown that the expression of OBPs can be altered depending on the insect’s diet. Indeed, the expression of one OBP of female *Culex nigripalpus* increased in the midgut, thorax and abdomen after a bloodmeal, suggesting a possible role in blood feeding [78,79]. Moreover, a diet change in *Anoplophora glabripennis* can affect gut-expressed OBPs together with other genes implicated in digestion, detoxification and nutrient acquisition. The feeding of *A. gabripennis* larvae in a host with documented resistance (*Populus tomentosa*) induced the downregulation of 5 OBP genes. It is not known whether alteration of the gut OBP gene expression is directly linked to the resistance of *A. gabripennis* to the *Populus tomentosa* plant [80]. Moreover, bacterial symbionts increase the gut expression of tsetse’s OBP6 and of OBP28a in *D. melanogaster* [81]. The roles of these OBPs are discussed in a later section of this review. Notably, nineteen *B. dorsalis* OBPs and seven *D. melanogaster* OBPs are highly expressed in the fat body [41] (Figure 2c), although their roles in the fat body remain unclear. It is important to acknowledge that the expression of OBPs in specific organs does not represent a proof of function. Further physiological studies are needed to fully investigate the role(s) of OBPs in the different parts of the insect body. In the absence of selection against OBPs expression, some OBPs still become expressed despite having no obvious function. This phenomenon could lead to rapid evolution of novel functions.

## 3. Biochemical Properties of OBPs

### 3.1. Structure of Insect OBPs 

Insect OBPs are small soluble proteins classified based on the number of amino acid residues found in their primary structure: long-chain OBPs (~160 residues), medium-chain OBPs (~120 residues) and short-chain OBPs (~100 residues). The similarity between the amino acid sequences of OBPs from the same species is low (<10% identity). The protein sequences of insect OBPs include highly conserved cysteines with a specific number of amino acid residues (AAs) between them [82,83]. In all cases, there are three AAs between the second and the third cysteines and eight AAs between the fifth and the sixth cysteines. OBPs were initially described in Lepidoptera and were divided into five subfamilies based on their amino acid sequences and tissue expression, providing a putative function: one pheromone-binding protein family (PBPs), two general odorant-binding protein families (GOBP1 and GOBP2) and two antennal binding protein families (ABP1 and ABP2), also called ABPx. The first OBP identified in the giant moth was called “PBP” based on its ability to bind radioactive pheromones [11]. This OBP was followed by the identification and cloning of the full-length cDNA sequence from (i) the tobacco hawk moth *Manduca sexta* PBP (MsexPBP) [84], (ii) the wild silkmoth *A. polyphemus* [85], and (iii) the Chinese oak silkmoth *Antheraea pernyi* [83]. GOBPs were detected in both male and female antennae of the tobacco hawk moth. More precisely, they were localized in basiconic sensilla, which respond to food odors. GOBPs are separated into GOBP1 and GOBP2 subfamilies on the basis of their amino acid sequences [86,87]. GOBPs are associated with general odorant-sensitive neurons. The two ABPx subfamilies are highly expressed in the *Bombyx mori* antennae and share some structural features with PBPs and GOBPs. However, there is no correlation between the ABPx sequences and PBPs or GOBPs [88,89]. In Diptera, OBPs have been classified into five structural groups depending on the number of conserved cysteines: (1) classic OBPs with the typical six-cysteine signature, (2) dimer OBPs containing two six-cysteine signatures, (3) plus-C OBPs with two additional conserved cysteines plus one proline, (4) minus-C OBPs that have lost two conserved cysteines and (5) atypical OBPs with 9–10 cysteines and a long C-terminus [90,91,92,93]. 

The three-dimensional structure of classic OBPs consists of a six α-helical domain forming a hydrophobic cavity [94] (Figure 4). The structural stability of insect OBPs depends on the presence of three interlocked disulphide bridges linking conserved cysteines [95,96,97]. Although the AA sequences of insect OBPs are highly divergent between and within species, the structure of insect OBPs is highly conserved. To date, crystal or NMR structures of more than 20 OBPs or PBPs from species belonging to different insect orders have been solved and are available in Entrez’s 3-D structure database at NCBI, accompanied by more than 20 detailed papers [94,98,99,100,101,102,103,104,105]. OBP structures have been solved in ligand-free (apo) or ligand-bound states, allowing researchers to study the interaction of the binding cavity with pheromones or with general odorants (Figure 4). The crystal structure of OBP1 from *Anopheles gambiae* and *Aedes aegypti* revealed a dimer with a unique binding pocket consisting of a continuous tunnel running through both subunits of the dimer [106,107]. 

### 3.2. Binding Properties of Insect OBPs

The binding properties of insect OBPs have been characterized using different techniques. Fluorescent binding assays showed that two PBPs (from *Antherea polyphemus* and *Mamestra brassicae*) are able to bind several pheromonal compounds, fatty acids (FAs) and long-chain alcohols [108]. Using the same approach, the capacity of *D. melanogaster* LUSH OBP to bind bulky and aromatic compounds, such as dibutyl phthalate was identified [93]. Similarly, the capacity of Drosophila OBP28a and OBP19b to bind floral-like chemicals and amino acids, respectively, was identified with the help of a competitive binding assay [37,94], which is the method of choice to study OBP-binding properties. The affinity of insect OBPs for odorants has been measured by isothermal titration calorimetry [97,109]. In addition, a tryptophan fluorescence quenching assay also revealed that LUSH OBP can bind the male pheromone *cis*-Vaccenyl acetate (cVA):cVA quenches LUSH Trp 123 in a cVA-concentration-dependent, saturable manner [110,111]. Notably, the X-ray crystal structure of LUSH bound to the cVA pheromone was solved, revealing that the “cVA–LUSH” interaction induces a specific conformational change of amino acid residues in the C-terminal region. The amino acid shifts in the C-terminal region induce the disruption of a salt bridge normally found in both the alcohol-bound and apo-LUSH structures [111]. The variation in the length of the C-terminus between insect OBPs affects ligand-binding mechanisms [112]. More precisely, a long C-terminus segment can enter the binding pocket, as in *B. mori* PBP1 [113], while a medium-length C-terminus covers the entrance to the binding pocket, as in the honeybee *Apis mellifera* PBP1 [100]. However, in a short C-terminus, such as the cockroach *Leucophaea maderae* PBP, the binding pocket is open to the external environment [29,114]. 

Other studies have described pH-dependent conformational changes during OBP binding [115,116,117]. This phenomenon was first observed in *Bombyx mori* PBP1 [113] and subsequently in other insect OBPs [106,107,118,119,120]. Lepidoptera PBPs possess a C-terminal region that is long enough to form a new helix. C-terminal non-polar amino acids undergo a histidine protonation switch at low pH that stabilizes the insertion of the new helix into the binding cavity. The C-terminal helix inside the pheromone binding site can compete with potential ligands. Ligand binding is only possible when the histidine residues are deprotonated at neutral pH, which leads to the extrusion of the unstructured C-terminus and exposure of hydrophobic residues of the binding sites. While Diptera OBPs undergo a pH-dependent conformational change leading to the loss of binding affinity, their C-terminus region is not long enough to form a new helix, which is why Diptera OBPs exhibit an alternative mechanism in which the C-terminal region acts as a “lid” covering the binding cavity. The stability of the “lid” is maintained by pH-sensitive hydrogen bonds. The ligand is only released from the OBP-odorant complex when the hydrogen bonds are disrupted in proximity to the dendritic membrane, where the pH is low [98,106,107,121] (Figure 5). Moreover, other OBP-binding mechanisms have been identified. For instance, at pH 4.0, *Apis mellifera* ASP1 exhibits a higher affinity to a main component of the queen bee pheromone than at pH 7.0 [103]. At pH 7.0, ASP1 is thought to undergo dimerization, which causes it to bind its ligand with a lower affinity compared with the acidic ASP1 monomeric form [104]. Other studies showed that the interaction of *D. melanogaster* LUSH OBP with sensory neuron membrane protein 1 (SNMP1) triggers ligand release. Ionization of the SNMP1 ectodomain may change the local pH, leading to conformational changes of LUSH OBP and the passage of ligands to SNMPs [121,122,123]. In vitro binding studies identify possible ligands to OBPs. The physiological role of OBPs in the perception of the identified ligands should be further investigated using behavioral assays and electrophysiology.

## 4. Diverse Chemosensory Functions of OBPs

Since the discovery of OBPs, several hypotheses and models have been proposed concerning their roles in chemoreception [125]. Later, studies using structural analyses, in vitro binding, behavioral assays and electrophysiological recordings revealed unsuspected roles of insect OBPs (see Table 2). 

### 4.1. Odorant and Pheromone Transport to Olfactory Receptors 

The relative low affinity of OBPs for odorants and pheromones, together with their high abundance in the sensillar lymph, led to the proposal that their roles consisted of binding, solubilizing and transporting hydrophobic stimuli to the chemoreceptors across the aqueous sensilla lymph. Studies carried out with varied insect species showed that OBPs are involved in the discrimination of odorants and oviposition attractants [126,127,128] and in the modulation of OR responses [129,130,131,132]. For example, the knockdown of the mosquito OBP CquiOBP1 alters adult electrophysiological responses to peculiar oviposition attractants [127]. Similarly, AgamOBP1 is involved in the intensity of odorant responses in *Anopheles gambiae* [126]. Knockdown of two *Aedes albopictus* OBPs, AalbOBP37 and AalbOBP39, altered the adult electrophysiological and behavioral responses towards indole which is an indicator of human sweat and breeding sites [128]. Further in vitro experiments demonstrated the role of OBPs in the solubilization of semio-chemicals. Such experiments consisted in monitoring the responses of OR-expressing cells exposed to different pheromones presented to heterologously expressed PBPs [129,130,131,132]. In Drosophila, the peri-receptor cascade of events led to the detection of the cVA pheromone. LUSH, also known as OBP76a, has been deeply investigated. LUSH solubilizes and transports cVA to OR67d, the cVA-dedicated receptor [111]. The cVA–LUSH interaction was proposed to induce a conformational change, triggering a specific binding of the “LUSH–cVA” complex to OR67d [111]. Moreover, a Drosophila CD36 homologue, sensory neuron membrane protein 1 (SNMP1), expressed in pheromone-sensing neurons is required for cVA detection. However, in vivo co-immunoprecipitation or cell culture surface-binding assays failed to provide evidence for SNMP1/LUSH complexes. While these data cannot exclude an interaction between SNMP1 and LUSH and they rather suggest that these proteins do not form a stable complex. Other reports present evidence that contradicts the proposed model in which the conformationally activated LUSH upon cVA binding interacts with the pheromone receptors. These studies described OR responses to pheromones in the presence of SNMP1 but without the relevant OBP [133,134,135], thus suggesting that pheromones alone are able to bind directly to SNMP1. Other studies showed that at high concentration, pheromones can directly induce OR-dependent responses in heterologous neurons or other cells in the absence of SNMP1 [136,137], leading to the idea that SNMP1 is not an integral part of the molecular machinery required for OSN firing. The precise biochemical mechanism of cVA detection remains unclear. These studies led to a proposed model explaining the mechanism of pheromone detection: in a pheromone-rich environment, cVA enter the lymph and is thought to be encapsulated by LUSH, which undergoes a conformational change. Subsequently, direct or indirect interaction of “cVA/LUSH” with SNMP1 induces the release and transfer of cVA to the ligand-binding site within the OR67d/ORCO complex. The biochemistry of the interaction between ORs and ligands is still unknown, but several reports suggest that the binding site lies within the transmembrane regions [138]. The presence of a central cavity in SNMP1 might be responsible of the delivery of cVA to the binding pocket [121,122] (Figure 6).

The OBP69a in *D. melanogaster* is also thought to be involved in the machinery modulating the behavioral responses to cVA. OBP69a shows a sexually dimorphic expression in fruit flies and is reciprocally regulated between male and female flies reared in similar social conditions. Exposure of flies to cVA was sufficient to decrease OBP69a expression in male flies and increase its level in female flies. The expression of OBP69a is regulated via a mechanism that depends on relaying the information from the sensory neurons to the second order olfactory neurons in the brain, and eventually back to OBP69a producing cells. OBP69a levels regulate the rate of aggressive displays in male flies in which down-regulation decreases—and up-regulation increases—aggressive behavior in single male flies. OBP69a promotes receptivity in response to cVA exposure in female flies [139]. A large-scale study using RNAi knockdown of OBPs induced decreased behavioral responses of *D. melanogaster* flies to a variety of odorants [140]. In *A. gambiae*, RNAi knockdown of OBPs affected electroantennogram responses to oviposition attractants [126]. However, the exact roles of these OBPs in the detection of pheromones and general odorants remain unknown.

### 4.2. Modulation of Mating Behaviour

It has been shown that OBP56h modulates *D. melanogaster* mating behaviour [141]. RNAi-mediated reduction in the expression of OBP56h alters the biosynthesis of cuticular pheromones, including the 5-tricosene (5-T) sex pheromone, which leads to the delay of copulation latency. More precisely, inhibition of OBP56h induces changes in the expression levels of genes associated with the gene ontology terms of lipase, triglyceride lipase activity, and phospholipase activity which are precursors of insect cuticular hydrocarbons [136,142]. 5-T is highly produced by males and in small quantities by females. The level of this pheromone was correlated with the delay to initiate male courtship in *D. melanogaster* and might therefore also decrease the probability of male–male courtship in nature [143,144,145]. The reduction of the 5-T amount enhanced mating frequency, likely by reducing courtship latency [141]. 

### 4.3. Sensitivity Modulation

Recent studies have highlighted the involvement of OBPs in the sensitivity of flies to odorants and sex pheromones. In Drosophila, deletion of OBP28a in the ab8 sensillum (OBP28a is the only OBP expressed in these sensilla) induced increased electrophysiological responses in different odorants tested over a broad concentration range. These data suggest that OBP28a acts as a buffer against sudden changes in odorant levels, which means that, after a sudden influx of odorant into the sensillum, OBP28a binds some of the odorant molecules to reduce the amount remaining available to activate ORs [34] (Figure 7a). However, the deletion of the abundant OBPs expressed in other basiconic sensilla (with the exception of ab4) did not affect their electrophysiological responses towards a wide variety of olfactory stimuli [146] (Figure 7b). The ab4 sensillum mutation elicited a stronger electrophysiological response and a lower threshold in oviposition preference towards linoleic acid compared to control flies [146] (Figure 7c). This finding indicates that OBPs also have a modulating effect on the olfactory physiology and on behaviour towards specific odorants.

Further investigation of OBP28a implicated its role in the detection of the floral odorant β-ionone [94]. Y-olfactometer assays revealed that the locomotion and the choice responses of OBP28a mutant flies were only altered at certain β-ionone concentrations. More precisely, control flies showed both higher locomotion to the choice point and a higher preference for the olfactometer arm containing 0.01 and 0.05 mM β-ionone compared to mutant flies. However, the responses of control and mutant flies to 1 mM β-ionone were not different (Figure 8a), indicating that mutant flies have decreased sensitivity to lower concentrations of β-ionone. Moreover, the ab4 and pb2 sensilla of mutant flies showed decreased electrophysiological responses to the highest β-ionone concentrations tested when compared to control flies (Figure 8b). These results indicate that the OBP28a deletion induced an increased threshold of the β-ionone detection [94]. The enhanced sensitivity role of OBP discovered in flies was supported with a *Bombyx mori* study. More precisely, BmPBP1-knockout males showed a reduced electrophysiological antennal response to bombykol (female sex pheromone) than wild-type males. The initiation of the orientation behaviour to the pheromonal source was also reduced in BmPBP1-knockout males [19].

OBPs could also participate in the termination of odorant response. In particular, OBPs might collaborate with esterase enzymes to inactivate the *A. polyphemus* sex pheromone after its interaction with the receptor [11,147]. In *D. melanogaster*, double deletion of OBP83a and OBP83b alters the deactivation kinetics towards some odorants, but do not have an effect on the activation kinetics. The odor-induced electrophysiological responses from the 10 potentially affected olfactory neurons in wild type and OBP83a and OBP83b mutants for the best-known activating ligands for each neuron were compared. The post-stimulus spiking activity of Or83c-, Or47b- and Or67d-expressing neurons stimulated with farnesol, trans-2-hexenal and cVA, respectively, persisted much longer in the OBP83a and OBP83b mutants than in controls [148].

### 4.4. Humidity Detection

OBPs are also involved in hygro-reception [57]. The genetic suppression of OBP59a expressed in the second chamber of the antennal sacculus (Figure 2b) affects Drosophila hygrotaxis. The preference of flies presented to a binary choice of high or low humidity was measured over different time scales. While control flies chose the humid sector, OBP59a-deficient mutant flies preferred the drier sector. This experiment indicates that OBP59a is involved in humidity perception (Figure 9a). Mutant flies also showed a reduced proboscis extension response (PER) to water vapor and, more unexpectedly, higher resistance to desiccation than control flies (Figure 9b) [57]. The molecular pathway of humidity detection by OBP59a is still unknown.

### 4.5. Haematopoiesis Modulator

The insect immune system largely depends on the symbiotic bacteria present in the gut. As indicated above, tsetse flies (Glossina spp.) host the maternally transmitted symbiont *Wigglesworthia*, which upregulates the expression of OBP6 in the gut of tsetse larvae. The transcript abundances of OBP6 and the hematopoietic RUNX transcription factor lozenge in tsetse embryos prior to and post maternal inoculation with siRNA were quantified and compared. The absence of OBP6 and lozenge transcripts during embryonic development after siRNA inoculation led to a dysfunctional melanization cascade during adulthood. Indeed, OBP6 is necessary for the formation of crystal cells, which induce the production of melanin during immune responses. The orthologous protein of OBP6 in tsetse is OBP28a in *D. melanogaster*. The reduction of OBP28 expression by RNAi also disrupts the melanization process. These data reveal the evolutionary conserved role of OBP in the hematopoietic program of insects [81].

### 4.6. Attraction and Aversion to Gustatory Cues

As mentioned above, OBPs can also be expressed in Drosophila taste sensilla [35,38,39,47,149]. In particular, OBP19b, OBP49a, OBP57d and OBP57e are involved in taste perception. OBP57d and OBP57e are two *D. melanogaster* proteins expressed in the leg sensilla and are involved in the oviposition response to C6–C9 fatty acids. Flies knocked down for either of these two OBPs showed an altered preference for the tested fatty acid compared with control flies [150]. Moreover, hybrids resulting from the cross between *D. melanogaster* deficient mutants and *D. sechellia* or *D. simulans* highlighted a shift of the oviposition site preference of *D. melanogaster* deficient mutants to that of *D. sechellia* or *D. simulans*, respectively. These results showed that the interspecies differences are, at least in part, controlled by the Obp57d/e genomic region, which also explains the specialization of *D. sechellia* (endemic to Seychelles islands) to the *Morinda citrifolia* toxic hostplant [149]. OBP49a, which is expressed in the *D. melanogaster* labellum, can suppress the appetence for sweet-tasting compounds through the perception of bitter stimuli. The deletion of OBP49a reduced the inhibition of sucrose-induced action potential by bitter chemicals [36] (Figure 10). The use of RNAi-mediated reduction of the expression of individual OBP genes induced either an increase or a decrease of sucrose intake in the presence of bitter compounds. While an increased intake suggests that OBPs transport bitter tastants to their cognate receptors and sequester the tastants, a decreased intake suggests a role of OBPs in the clearance of bitter tastants [151]. Moreover, OBPs could be involved in the perception of toxic compounds. For instance, OBP11, expressed in the basiconic sensilla of *Adelphocoris fasciaticolli* labellum, plays a crucial role in the detection of gossypol, a toxic secondary metabolite. Indeed, OBP11 showed high affinities to non-volatile compounds, including gossypol. The biological function of OBP11 was studied by measuring the total ingestion duration of insects using electrical penetration graph (EPG) tests. RNAi-mediated reduction of OBP11 expression led to an increase of the total ingestion time of insects on an artificial diet containing 2.0 % gossypol. These data suggest that the OBP11 is important for the sensitivity of heteropterus insects towards gossypol [152].

OBP19b was recently identified as a major factor involved in the detection of specific amino acids. Ligand binding assays revealed that OBP19b binds a subset of L-amino acids (Figure 11a). Drosophila mutants devoid of OBP19b showed an altered preference to these L-amino acids (L-phenylalanine and L-glutamine) compared to control flies (Figure 11b). Mutant flies also showed decreased electrophysiological responses of single-taste proboscis sensilla towards the same amino acids (Figure 11c). Given that the OBP19b-like protein coding sequence is highly conserved in various dipteran insects, it might play a critical role in the detection of amino acid-rich food [37]. Future studies should aim to better decipher the link between the peripheral and central nervous systems involved in amino acid perception [153]. Indeed, the Drosophila protein appetite is regulated by two central system regions: (i) a small cluster of dopaminergic neurons enhancing yeast intake in protein-deprived flies [154] and (ii) the protein-specific satiety hormone FIT, which inhibits protein-rich food intake [155]. In addition, study of the ability of the OBP19b three-dimensional structure to bind amino acids will help to solve the OBP–amino-acid interaction at the biochemical level.

### 4.7. Perspectives on Genetic Analysis of OBPs

While a number of OBPs have been mutated, most in Drosophila, the resulting phenotypes define the function of individual OBP members better than anything else. Some mutants have unexpected phenotypes that do not fit neatly into current models and this may open new paths of investigation. With CRISPR technology opening all species to genetics, study of OBP expression in moth or other non-model insect species receptors and binding protein genes will open a new era of functional analysis. Some studies have used Drosophila as a tool to deorphanize moth ORs and to investigate the functional interaction between PBPs and pheromone receptors. The implication of PBPs in the detection of (Z)-11-hexadecenal, a major sex pheromone *of Helicoverpa armigera*, was recently studied. HarmOR13, the primarily ORs responding to (Z)-11-hexadecenal and two PBPs (HarmPBP1, HarmPBP2) were heterologously expressed in Drosophila T1 sensilla. This report specially revealed that the response of HarmOR13 to the moth pheromone increased in the presence of HarmPBP1 or HarmPBP2. However, the selectivity and the response kinetics of HarmOR13 were not modulated by the presence of either HarmPBP1 or HarmPBP2 [156].

## 5. Conclusions

(1) Studies conducted in the last 40 years have provided information regarding the different roles of insect OBPs. OBPs were believed to be only expressed in olfactory organs and to be strictly involved in chemoreception mechanisms. However, an increasing number of reports has revealed that OBPs are expressed in most organs of the insect body and have non-conventional roles, including in taste, immunity response and humidity detection.

(2) We present the latest discoveries made on the structural and binding properties of insect OBPs. We focus on the properties of insect OBPs and, more specifically, on their tissue and cellular expression. We also present the varied functional roles, both classical and some non-conventional, of currently known OBPs.

(3) Yet there is massive amount of information on OBP functions that we ignore and needs to be investigated. These studies will pave the way to different technological applications in environmental, food quality and medical fields.

(4) As mentioned above several reports have shown that insects OBPs can modulate the response of ORs to odorants, nevertheless the molecular details of such mechanism remain unclear. Several options can be presented: (i) OBPs might release the odorants at the proximity of ORs leading to the formation of odorant-ORs complex and ORs activation, (ii) the complex odorant-OBPs might directly interact and activate ORs, (iii) OBPs buffer odorants in the lymph by limiting the number of odorant molecules available to activate the ORs.

(5) Insect OBPs and vertebrate OBPs (a large family of ligand-binding proteins, that belong to the lipocalin family) share similar stability and versatility properties, even thought that the two families of proteins are structurally distinct. The implication of OBPs in eliciting the behavioral response and coding of odor has mainly been demonstrated in insects. Even though the role of vertebrate OBPs at the level of the respiratory apparatus remains unclear, some reports showed the role of vertebrate OBPs as a protector against oxidative stress. Vertebrate OBPs scavenge highly reactive low molecular aldehydes and alken-aldehydes which produced in consequence of peroxidation of membrane unsaturated fatty acids [157,158]. Moreover, a recent study showed that vertebrate OBPs might behave as humoral components of innate immunity, active against pathogenic bacteria and fungi. Ligand binding assays showed that bovine and porcine forms of the Lipocalin OBPs bind to quorum sensing molecules of the bacterium *Pseudomonas aeruginosa* (PA) and the yeast *Candida albicans* (CA). The direct antimicrobial activity of the bovine and porcine OBPs against CA and PA was also revealed [159].

Other studies suggested that vertebrate OBPs are pheromone carriers in biological glands or secretary body fluids like urine, saliva, seminal fluid [13,160,161,162,163,164] and can also have a role in olfaction [164,165,166,167]. Moreover, a role in odor perception and sexual communication in buffaloes has been proposed [26,28], and a recent study investigated the binding with buffalo estrus-specific pheromones by fluorescence quenching assays and mutational studies [168]. All these studies further support the functional similarities between OBPs in insects and lipocalins in vertebrates.

(6) A diversity of ligand binding proteins exists in nature and have been engineered to design biosensors for specific detection of various biomolecules. The conformational changes caused by the ligand-binding are converted into electrical signals, magnetic responses or fluorescence that allow the biosensing of different disease markers, pathogenic molecules, environmental toxins and chemically or biologically hazardous compounds. Among these ligand binding proteins, the periplasmic binding proteins, found in bacteria and archaea, are involved in chemotaxis and solute uptake [169,170,171]. A large variety of periplasmic binding protein ligands including carbohydrates, AAs, anions, metal ions, dipeptides and oligopeptides were identified. Biosensors detecting AAs such as L-glutamine and L-leucine were successfully produced with such specific periplasmic binding proteins [172,173]. AAs are reliable indicators of the nutritive value of the food and could therefore be used to monitor many fermentation processes and to detect the presence of bacterial activity. L-phenylalanine is also used to diagnose phenylketonuria (PKU), a genetic disorder of phenylalanine metabolism [174]. Other biosensors for odors were also developed using vertebrates and insects OBPs [29,175]. A study made use of a mammalian OBP to remove the herbicide atrazine, a dangerous pollutant [176]. The pig OBP was also incorporated into the fabrics of clothes to remove the cigarette odor and to release pleasant fragrances bound to this OBP [177]. Recently, an in silico analysis of human OBP established the relationship between the physicochemical properties of the odorants and the type and strength of binding, which could be useful in the design of technological applications of aromas and biosensors [178]. Moreover, an in vitro assay was designed using *Anopheles gambiae* OBP (AgamOBP1) to evaluate the presence in water of indole, a characteristic metabolite of harmful coliform bacteria [179].

## Figures and Tables

**Figure 1 biomolecules-11-00509-f001:**
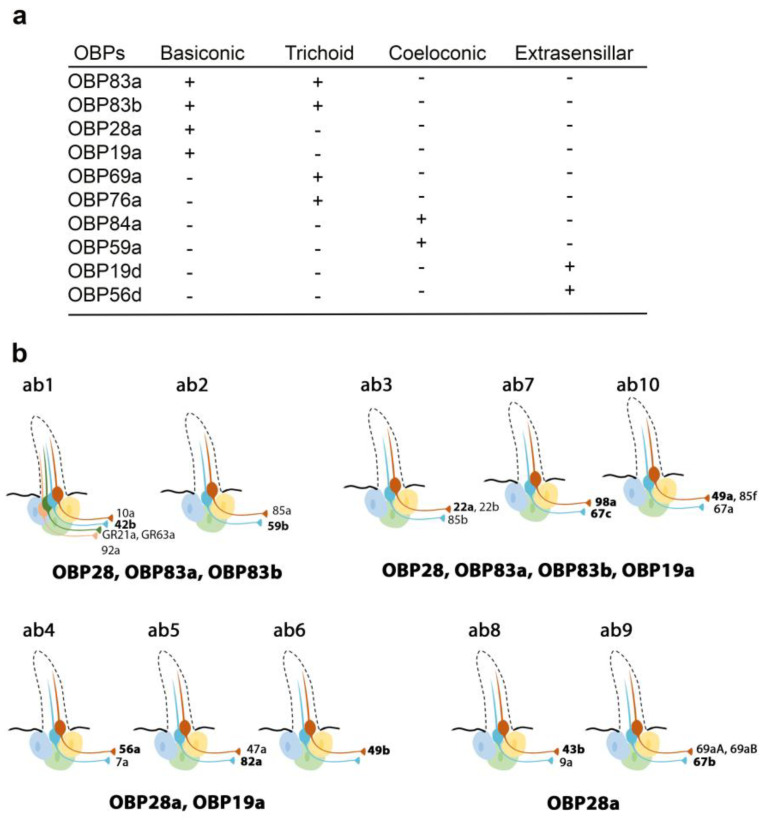
Expression patterns of the most abundant OBPs in Drosophila antennae. (**a**) Summary of the OBP expression patterns in the three types of antennal sensilla. OBP19d and OBP56g are expressed in epidermal cells. (**b**) Distribution of highly abundant OBPs expressed in Drosophila antennal basiconic sensilla. Olfactory receptor (OR) genes expressed in each olfactory receptor neuron class are indicated (all the OR genes except for GR21a and GR63a). The OR markers used for in situ hybridization for each sensilla are shown in bold [28,34].

**Figure 2 biomolecules-11-00509-f002:**
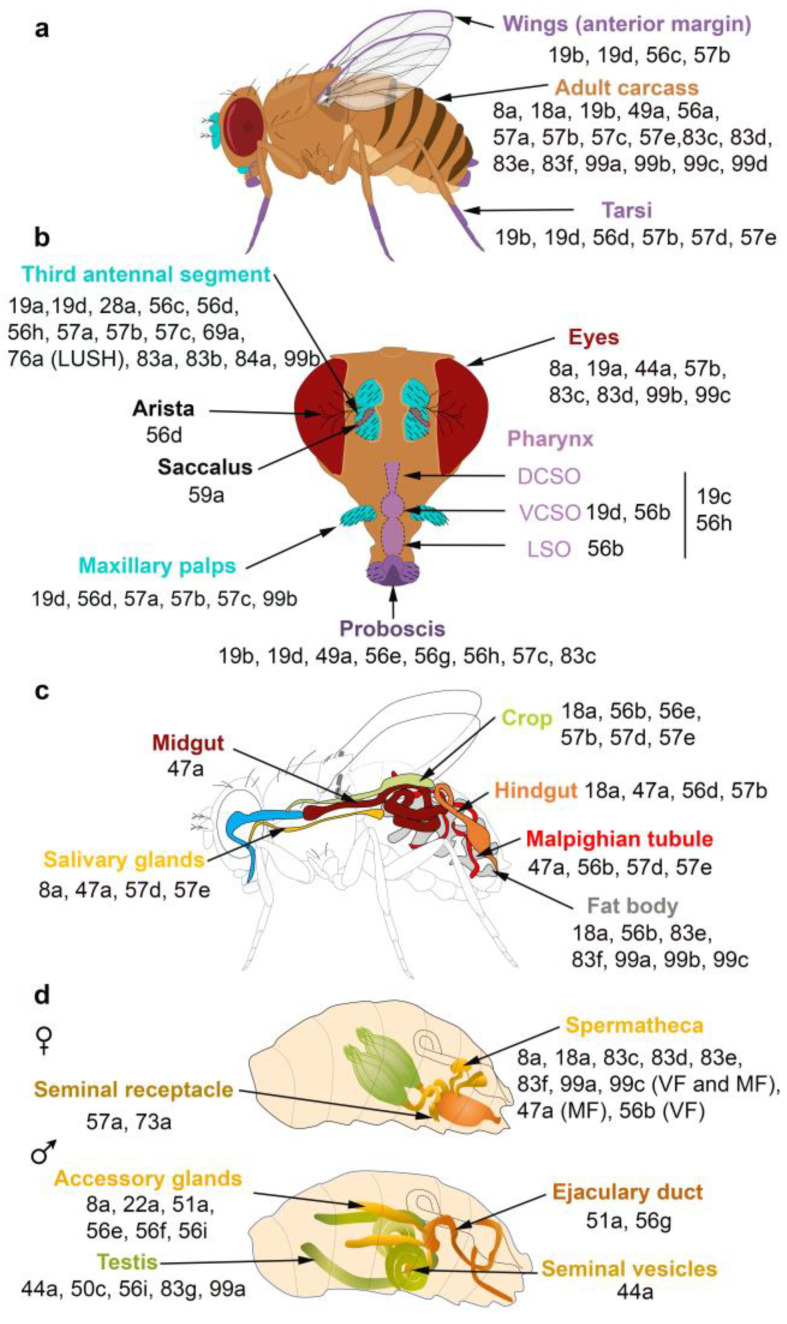
OBPs expressed in the olfactory, gustatory, digestive and reproductive organs of *D. melanogaster*. (**a**,**b**) Adult taste organs (purple) consist of the proboscis, the internal structure in the pharynx (the labral sense organ (LSO), the ventral and dorsal cibarial sense organs (VCSO and DCSO)), the leg tarsi, the anterior margin of the wings and the female genitalia. Adult olfactory organs (turquoise) consist of the antenna and the maxillary palps. OBPs expressed in each organ are indicated by their number. OBPs are found in chemosensory and non-chemosensory organs. (**c**) Expression patterns of OBPs in the adult digestive tract. (**d**) Expression patterns of OBPs in the female and male reproductive organs. The expression of OBPs in female spermatheca depends on the sexual state. Some OBPs are expressed in virgin female (VF) and mated female (MF) spermatheca, while the expression of some OBPs is mating-dependent [36,37,38,47,54,55,56,57,58,59].

**Figure 3 biomolecules-11-00509-f003:**
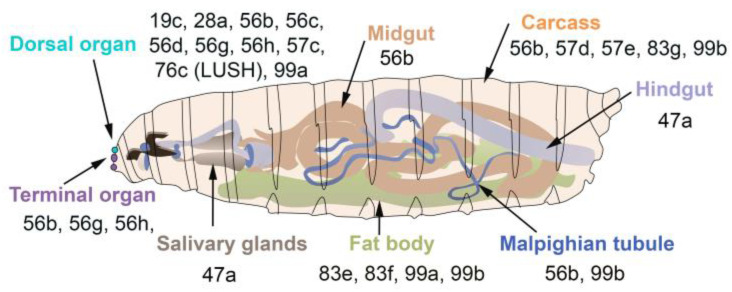
OBPs expressed in the olfactory gustatory and digestive organs of *D. melanogaster* larvae. The larval dorsal organ (turquoise) detects volatile odorants, while the larval terminal organ (purple) detects both soluble and volatile chemicals.

**Figure 4 biomolecules-11-00509-f004:**
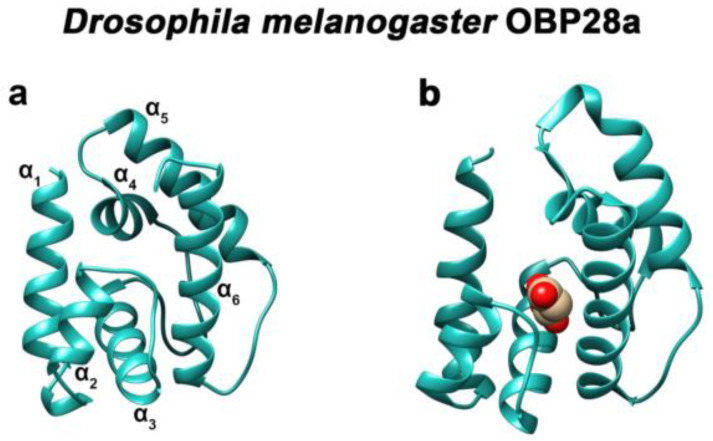
Three-dimensional structure of *D. melanogaster* OBP28a. (**a**) Apo- and (**b**) bound-three-dimensional structures of the fruit fly (Drosophila melanogaster) OBP28a (PDB: 6QQ4). α_1_ to α_6_ indicate the six α-helices found in the structure of classical insect OBPs. Drosophila OBP28a is in complex with penta-ethylene glycol [94].

**Figure 5 biomolecules-11-00509-f005:**
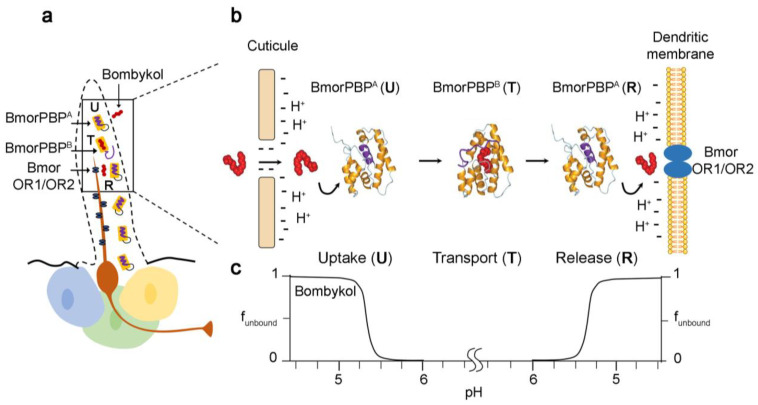
Model for the mechanism of *B. mori* pheromone transport depending on the pH-linked structural flexibility of BmorPBP. (**a**) Schematic representation of Bombykol sensing sensillum. The box highlights the three steps of pheromone detection: uptake by BmorPBP^A^ (U) leading to the conformational transition to BmorPBP^B^ and the transport of the pheromone to OR1/OR2 receptor complex (T) leading to the release (R) of the pheromone and to the conformational change to BmorPBP^A^. (**b**) Structural basis of *B. mori* pheromone transport. After entering the senislla, Bombykol bind to BmorPBP^A^ due to the reduced pH near the pore. The uptake leads to the conformational change of BmorPBP^A^ to BmorPBP^B^ and to the transport of the pheromone to the receptor complex. The acidic milieu near the membrane reduces the stability of the BmorPBP^B^–Bombykol complex and leads to the release of the pheromone to the receptor. (**c**) Graph representing the fraction of Bombykol not bound to BmorPBP (f_unbound_) depending on the pH profile in *B. mori* sensilla. Adapted from [124].

**Figure 6 biomolecules-11-00509-f006:**
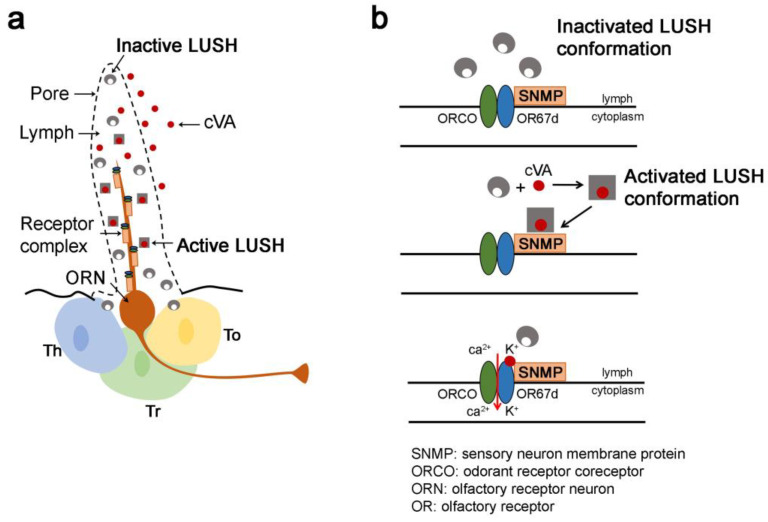
Model of the mechanism of the conformational activation of LUSH leading to *cis*-Vaccenyl Acetate (cVA) pheromone detection. (**a**) Schematic drawing of at1 cVA-sensing trichoid sensillum. This sensillum is punctured by multiple pores and houses one olfactory receptor neuron (ORN). The heteromeric complex composed of the OR67d pheromone receptor/ORCO coreceptor and the CD36-related sensory neuron membrane protein (SNMP) is localized on the ORN dendrite. Three accessory cells, trichogen (Tr), thecogen (Th) and tormogen (To) cells, surround the cell body of the ORN. Thecogen and the tormogen cells both secrete sensillar lymph components, including OBPs. (**b**) Top: in the absence of the cVA pheromone, LUSH is in an inactive state, and when bound to cVA, LUSH undergoes a conformational change. Middle and bottom: subsequently, direct or indirect interaction of “cVA/LUSH” with SNMP1 induces the release and transfer of cVA to the ligand-binding site within the OR67d/ORCO complex. Adapted from [121,122].

**Figure 7 biomolecules-11-00509-f007:**
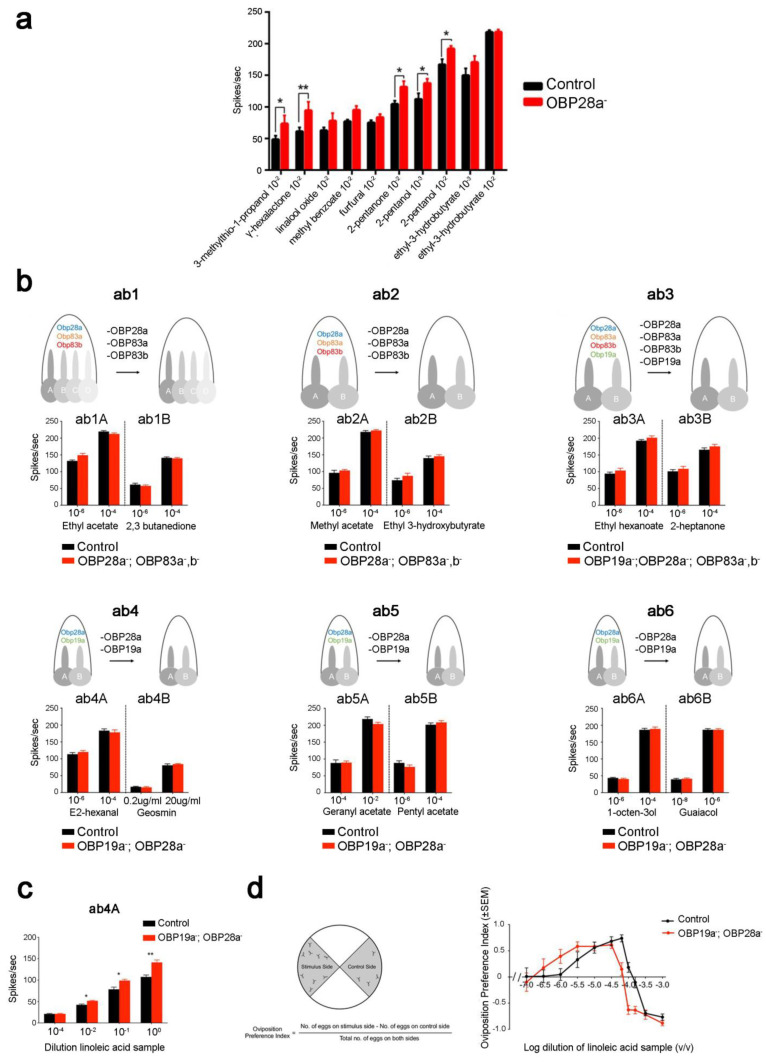
Role of OBP28a in the sensitivity of flies to general odorants. (**a**) Electrophysiological responses of ab8 neurons in control and OBP28a mutant flies to 0.5-s pulses of several chemical classes of odorants. Significant differences are indicated as *: *p* < 0.05, **: *p* < 0.01. (**b**) Electrophysiological responses of individual ORNs in ab1-ab6 sensilla of control and mutant flies. Mutant flies are deleted for the genes coding for abundant OBPs; their responses are shown to 0.5-s pulses of strong ligands. The deletion of abundant OBPs did not affect the mutant responses (Mann-Whitney U test). (**c**) Electrophysiological responses of ab4 sensilla towards linoleic acid in control and in [OBP19a^-^; OBP28a^-^] double mutant flies. Significant differences are indicated as *: *p* < 0.05, **: *p* < 0.01. (**d**) Oviposition preference of control and double mutant flies to linoleic acid. Left: Schematic illustration of the two-choice oviposition preference paradigm and response. Bottom: equation used to calculate the oviposition preference index. Right: graph representing the oviposition preference of control and mutant flies depending on the dilution of linoleic acid. Adapted from [34,146].

**Figure 8 biomolecules-11-00509-f008:**
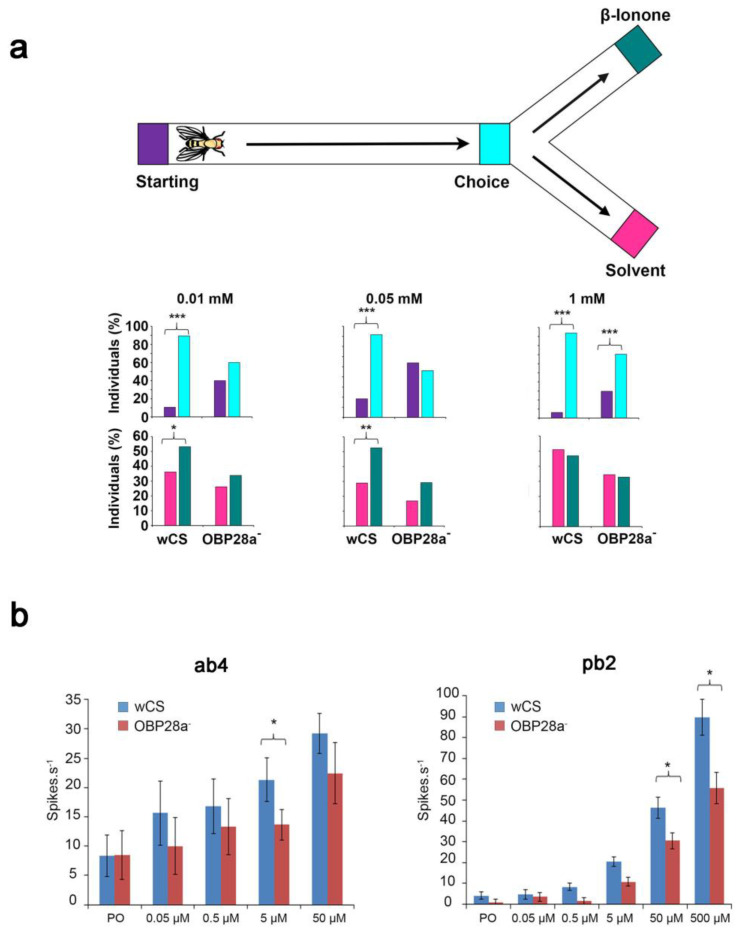
Deletion of OBP28a changes the threshold of detection of β-ionone. (**a**) Olfactory responses of control (wCS) and OBP28a mutant (OBP28a^-^) flies to three concentrations of β-ionone. Top: Illustration of the Y-shaped olfactometer used to test the fly olfactory preference for β-ionone. The position of the fly in the device is noted at two positions: first its locomotion in the straight tube (between the starting point (purple) and the choice point (blue)); second its preference relative to the arm chosen (either β-ionone (green) or the solvent (pink)). Bottom: percentages of control (wCS) and OBP28a mutant (OBP28a^-^) flies in each section of the device: the top histograms show the position in the starting tube and the bottom histograms show the arm chosen. *** *p* < 0.001; ** *p* < 0.01, Fisher test. (**b**) Electrophysiological responses of ab4 and pb2 sensillum of control (wCS) and OBP28a mutant (OBP28a^-^) flies to β-ionone in paraffin oil solvent. The responses are shown to 0.5-s pulses of increasing concentrations of β-ionone. * *p* < 0.05, ANOVA, Kruskal–Wallis, Dunn’s post hoc test. PO paraffin oil. Adapted from [94].

**Figure 9 biomolecules-11-00509-f009:**
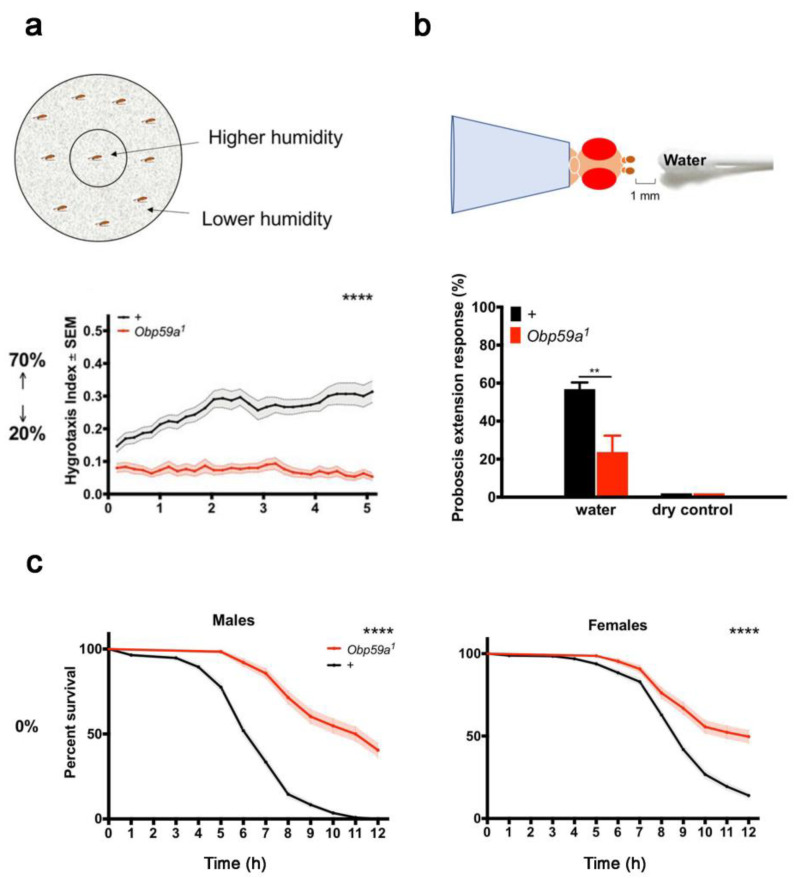
OBP59a is necessary for the detection of humidity and desiccation resistance. (**a**) Top: schematic representation of the hygrotaxis paradigm in a Petri dish. Bottom: Hygrotaxis responses of control (black) and OBP59a-deficient mutant (red) flies presented a choice between 20% and 70% humidity. **** *p* < 0.0001; Friedman test with Dunn’s multiple comparisons test. (**b**) Top: Illustration of the proboscis extension response paradigm (PER). Bottom: PER responses of control and OBP59a mutant flies to water vapor. ** *p* < 0.01; Mann-Whitney test. (**c**) Accumulated survival rate of control (black) and OBP59a mutant (red) male and female flies placed under desiccating conditions. **** *p* < 0.0001, log-rank (Mantel-Cox) test. Adapted from [57].

**Figure 10 biomolecules-11-00509-f010:**
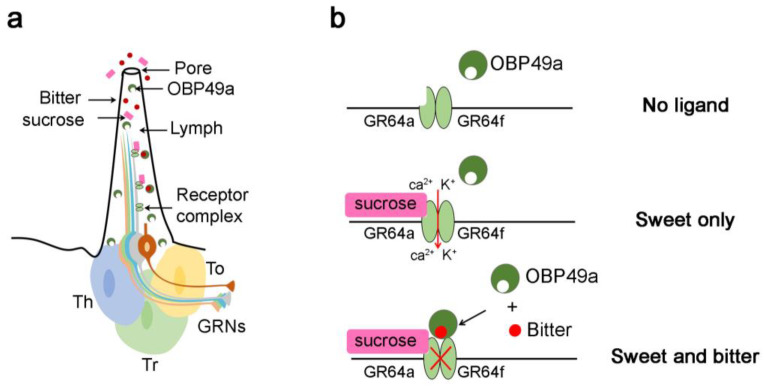
Hypothetical model of the suppression mechanism of sweet taste by bitter chemicals. (**a**) Schematic representation of sugar-sensing L-type sensilla. L-type sensilla end with a terminal pore and house four gustatory receptor neurons (GRNs) and one mechanosensory neuron. Three accessory cells, trichogen (Tr), thecogen (Th) and tormogen (To) cells, surround the cell body of each GRN. The GRN dendrites are bathed in the sensillum lymph containing OBP49a. (**b**) Schematic representation of the role of OBP49a. Top and middle: the sucrose receptor complex, GR64a with GR64f, is only active in the presence of sucrose. Activation of this complex leads to the entry of calcium and potassium into the neuron. Bottom: the presence of bitter compounds with sucrose inhibits the GR64a and GR64f receptor complex. More precisely, bitter compounds bind to OBP49a, and this complex inhibits the receptor even in the presence of sucrose. Adapted from [36,122].

**Figure 11 biomolecules-11-00509-f011:**
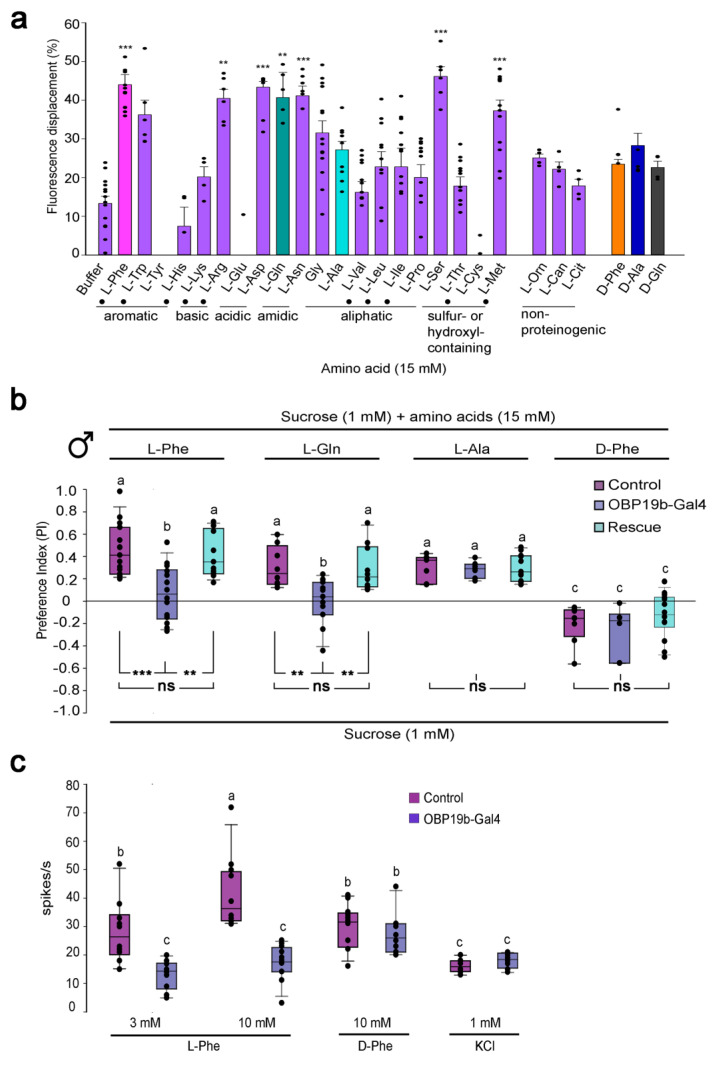
OBP19b is essential for amino acid detection in Drosophila. (**a**) OBP19b displays a high binding affinity for several amino acids. Graph representing the fluorescent displacements of 20 L-amino acids, three non-proteinogenic amino acids (L-ornithine (L-Orn), L- canavanine (L-Can), L-citrulline (L-Cit)) and three D-amino acids (D-phenylalanine (D-Phe), D-glutamine (D-Gln) and D-alanine (D-Ala). The AAs tested are shown in colored bars: three L-amino acids (L-phenylalanine, L-Phe: magenta color, L-glutamine, L-Gln: green and L-alanine, L-Ala: cyan) and three D-amino acids (D-Phe: orange, D-Gln: blue, and D-Ala: grey). The dots indicate essential amino acids. *** *p* < 0.001; ** *p* < 0.01; one-way ANOVA followed by Dunnett’s test. (**b**) OBP19b deletion affects Drosophila amino acid taste preference. Preference indices of control, OBP19b-Gal4 null mutant and genetically rescued male flies (OBP19b-Gal4: UAS-OBP19b) for two OBP19b ligands (L-Phe and L-Gln) and two non-ligands (L-Ala and D-Phe). The letters indicate significant differences determined by the Kruskal–Wallis test and a post hoc Wilcoxon test: ns: non-significant, ** *p* < 0.01; and *** *p* < 0.001. (**c**) OBP19b deletion alters the single proboscis sensilla electrophysiological responses of Drosophila. Mean ± SEM for the number of spikes/s obtained in control and mutant flies stimulated by 3 and 10 mM L-Phe, 10 mM D-Phe and 1 mM KCl. a/b or b/c: *p* < 0.01; a/c: *p* < 0.001, ANOVA and Tukey’s post hoc test. Adapted from [37].

**Table 1 biomolecules-11-00509-t001:** Numbers of annotated odorant-binding proteins (OBP) genes in different insects [27,28,29].

	Species	OBP Gene Number
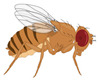	*D. melanogaster*	52
*D. simulans*	52
*D. sechellia*	51
*D. yakuba*	55
*D. erecta*	50
*D. ananassae*	50
*D. pseudoobscura*	45
*D. persimilis*	45
*D. willistoni*	62
*D. mojavensis*	43
*D. virilis*	41
*D. grimshaw*	46
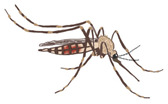	*Anophele gambiae*	69
*Culex quinquefasciatus*	109
*Aedes aegypti*	111
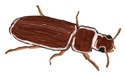	*Tribolium castaneum*	49
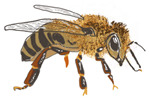	*Apis mellifera*	21
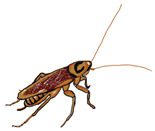	*Blatella germanica*	109
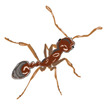	*Solenopsis invicta*	18
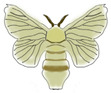	*Bombyx mori*	44

**Table 2 biomolecules-11-00509-t002:** The different roles of the OBPs in *Drosophila melanogaster*.

OBP	Role	Publication
OBP76a (LUSH)	Solubilization, transport and interaction with SNMP1	[111,122,134]
OBP69a	Implication in cVA response, role remains unclear	[139]
OBP28a	Modulation of olfactory sensitivity	[34,94]
OBP59a	Humidity detection	[57]
OBP57d and OBP57e	Modulation of oviposition site preference to C6-C9 acids in *D. melanogaster* and *D. sechellia* Specialization of *D. sechellia* to its host plant (*Tahitian Noni*)	[149] [150]
OBP49a	Suppression of the appetence for sweet compounds through the perception of bitter chemicals	[36]
OBP56h	Modulation of mating behaviour by alteration of cuticular hydrocarbon profiles in males	[141]
OBP19b	Detection of peculiar amino acids	[37]

## Data Availability

Not applicable.

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
