# Peer review of "The 40-Year Mystery of Insect Odorant-Binding Proteins"

_biomolecules, 2021, doi:10.3390/biom11040509_

Round 1

Reviewer 1 Report

This is a very rich review on many aspects and roles of odorant-binding proteins (OBPs) in insects. In recent years we have witnessed a proliferation of studies on insect olfaction and specifically on transport proteins, most on OBPs.

Although other reviews have been published even in the recent past, this fast-growing field requires frequent reports updating the current information with new developments.

This review perform this task timely and with competence. The amount of information reported is huge, very accurate and clearly presented.

Although covering all general aspects of insect OBPs, this review interestingly expands two fields that specifically need to be reviewed: chemoreception in Drosophila, because of its very rapid expansion, and the role of OBPs in taste, a very important aspect that has been unduly neglected for too long.

The well written text and the good figures make this manuscript a pleasure to read.

I have no special comment to make, except for congratulating with the authors for this excellent review of the literature, that I am sure will represent an essential reference for all those working with insect OBPs and particularly for the Drosophila scientific community.

I have only a couple of very minor points:

  1. Line 42: Cotton => cotton
  2. Line 142: the reference to the OBP of H. armigera is just a repetition of what reported on line 141 (as it is written, it looks like the authors refer toan OBP9 in H. armigera)
  3. Line 162: Peiletier => Pelletier; besides, the two references are in a different style (names instead of numbers)
  4. Line 175: Armigera => armigera
  5. Line 331: the ref should be [102] rather than [124]. Perhaps the all sentence could be rephrased to make clear what is stated in ref [102] and what is confuted in ref [124]
  6. Line 479: it could better be “tsetse’s OBP6

Author Response

We wish to thank the reviewer #1 and the editor for their insightful comments, which significantly helped us to improve our manuscript.

The authors agree with the reviewer’s #1 comments and included all the minor points suggested.

Line 42: Cotton => cotton : We corrected the mistake.

Line 142: the reference to the OBP of H. armigera is just a repetition of what reported on line 141 (as it is written, it looks like the authors refer to an OBP9 in H. armigera): We corrected the mistake

Line 162: Peiletier => Pelletier; besides, the two references are in a different style (names instead of numbers): We corrected the mistake.

Line 175: Armigera => armigera: We corrected the mistake.

Line 331: the ref should be [102] rather than [124]. Perhaps the all sentence could be rephrased to make clear what is stated in ref [102] and what is confuted in ref [124]. We corrected the mistake and rephrased the sentence.

Line 479: it could better be “tsetse’s OBP6”: We corrected the mistake.

Reviewer 2 Report

The work “Rihani at all” on OBP is quite interesting to read and I do favorably support their publication. The manuscript provides in deep view on the role of insects OBP. But authors mention OBPs in general, but the paper is just about insect OBP, therefore I would suggest a comparison of structural and binding characteristics with mammals OBPs (a small paragraph is enough since the title of the paper mention OBPs in general)

For that I think the authors could mention the recent papers:

  • The structural properties of odorants modulate their association to human odorant binding protein, Biomolecules, 11(2), 145, 2021(https://doi.org/10.3390/biom11020145)
  • Biotechnological applications of mammalian odorant-binding proteins, Crit. Rev. Biotech, 2021, https://doi.org/10.1080/07388551.2020.1853672

Author Response

We wish to thank the reviewer #2 and the editor for their insightful comments, which significantly helped us to improve our manuscript.

The authors agree with the reviewer’s #2 comments and modified the title to “The 40-year mystery of insect odorant-binding proteins”.

Following the referee’s remark, we mentioned in our discussion two recent papers:

-The structural properties of odorants modulate their association to human odorant binding protein, Biomolecules, 11(2), 145, 2021(https://doi.org/10.3390/biom11020145).

-Biotechnological applications of mammalian odorant-binding proteins, Crit. Rev. Biotech, 2021, (https://doi.org/10.1080/07388551.2020.1853672).

Reviewer 3 Report

The review by Rihani et al covers the structure, binding properties, physiologic roles and expression patterns of invertebrate OBPs. I have a few suggestions that might strengthen the review.

The review really focuses on invertebrate OBPs and really does not dwell at all on lipocalin vertebrate members, so perhaps adding invertebrate to the title would be appropriate, or alternatively, including a more detailed discussion of the vertebrate members. Some have been implicated as pheromones.

The review is divided into 2  major sections, Expression Patterns and Function.

For expression patterns, it is important to acknowledge that because a protein is expressed in a specific sensillum, that does not prove it functions there. If it does no harm but has no function, there is no selection against its expression. Indeed this may be key in the rapid evolution of novel functions. Similarly, in vitro binding of ligands to OBPs discussed in the function section does not correlate well with biological roles for the most part, unless specifically examining known activating ligands.

Since gene numbers among species is discussed, I think the authors should consider a discussion of the evolution of this gene family, perhaps relating this to ecological niche. The rapid evolution and diversity of this family is fascinating.

For the Function section, I am not sure ‘structure’ and ‘binding properties’ are the best subheadings. Perhaps make these their own topic sections after expression. As far as function goes, I strongly recommend including a section on the Genetic Analysis of OBPs. A number of OBPs have been mutated, most in Drosophila so far, and the resulting phenotypes define the function of individual OBP members better than anything else. Some mutants have unexpected phenotypes that do not fit neatly into current models that may open new paths of investigation. Similarly, expression of moth or other insect species receptors and binding protein genes in Drosophila is opening a new era of functional analysis (eg. see Guo et al. Insect Biochemistry and Molecular Biology, 131 (2021) 103554). With CRISPR opening all species to genetics, a section on Genetic Analysis is a worthy addition.

Author Response

We wish to thank the reviewer #3 and the editor for their insightful comments, which significantly helped us to improve our manuscript.

The authors agree with the reviewer’s #3 comments and therefore modified the title to “The 40-year mystery of insect odorant-binding proteins” (as mentioned above).

As suggested by the referee and to support the idea that the expression of OBPs in specific organs do not represent a proof of function, we added in the section “Tissue expression and cellular localization of OBPs” the following text: “It is important to acknowledge that the expression of OBPs in specific organs do not represent a proof of function. Further physiological studies are needed to fully investigate the role(s) of OBPs in the different parts of the insect body. In the absence of selection against OBPs expression, some OBPs still get expressed despite having no obvious function. This phenomenon could lead to rapid evolution of novel functions.

As suggested by the referee and to support the idea that in vitro binding results are not sufficient to prove a function of OBPs, in the section “Binding properties of insect OBPs” we added the following text:” in vitro binding studies identify possible ligands to OBPs. The physiological role of OBPs in the perception of the identified ligands should be further investigated using behavioural assays and electrophysiology.”

The authors agree with the reviewer’s #2 comment concerning the interesting aspect of the evolution of OBP gene family, and therefore included a paragraph on this matter in the section “Expression pattern of insect OBPs”: “Exhaustive comparative genomic analysis of OBPs gene families in 20 Arthropoda species revealed a highly dynamic evolution, with a high number of gains and losses of genes. The number of members of OBPs is variable and diverse across Arthropoda species, exhibiting a wide range of gene lengths and encoding different cysteine profiles. Interestingly, two OBP members (OBP73a and OBP59a) have clear orthology relationships not only in the 12 Drosophila genomes but also in almost all insect species (except in Hymenoptera). Studies in the organization in chromosome clusters of OBP genes showed that this family of genes is significantly clustered across the Drosophila evolution. This conservation across 400 My of evolution suggests the existence of some functional constraints maintaining the clusters [30]. Other reports revealed that OBPs were only present in the Hexapoda (insects), and absent in other arthropod subphyla including the nonhexapod pancrustaceans, chelicerates and myriapods. Moreover, OBPs genes were detected in basal hexapods, such as Archaeognatha, Zygentoma, and Phasmatodea. However, the origin of OBPs genes is still unknown and need further investigations [31,32].”

As suggested by the referee, the authors also rearranged the sections and separated the sections on “Structure of insect OBPs” and on “Binding properties of insect OBPs” from the section on “Diverse chemosensory functions of OBPs”.

As suggested by the referee, we added a section on the genetic analysis of OBPs in the Discussion and mentioned the suggested paper: “While a number of OBPs have been mutated, most in Drosophila, the resulting phenotypes define the function of individual OBP members better than anything else. Some mutants have unexpected phenotypes that do not fit neatly into current models and this may open new paths of investigation. With CRISPR technology opening all species to genetics, study of OBP expression in moth or other non-model insect species receptors and binding protein genes will open a new era of functional analysis. Some studies have used Drosophila as a tool to deorphanize moth ORs and to investigate the functional interaction between PBPs and pheromone receptors. The implication of PBPs in the detection of (Z)-11-hexadecenal, a major sex pheromone of Helicoverpa armigera, was recently studied. HarmOR13, the primarily ORs responding to (Z)-11-hexadecenal and two PBPs (HarmPBP1, HarmPBP2) were heterologously expressed in Drosophila T1 sensilla. This report specially revealed that the response of HarmOR13 to the moth pheromone increased in the presence of HarmPBP1 or HarmPBP2. However, the selectivity and the response kinetics of HarmOR13 were not modulated by the presence of either HarmPBP1 or HarmPBP2 [158].”

Round 2

Reviewer 2 Report

the paper is fully acceptable now